# Synergistic Repellent and Irritant Effects of a Mixture of β-Caryophyllene Oxide and Vetiver Oil against Mosquito Vectors

**DOI:** 10.3390/insects14090773

**Published:** 2023-09-20

**Authors:** Jirod Nararak, Unchalee Sanguanpong, Chutipong Sukkanon, Sylvie Manguin, Theeraphap Chareonviriyaphap

**Affiliations:** 1Department of Entomology, Faculty of Agriculture, Kasetsart University, Bangkok 10900, Thailand; fagrjrn@ku.ac.th; 2Association of Thai Innovation and Invention Promotion, Prachatipat 12130, Thailand; atipcontact@gmail.com; 3Department of Clinical Microscopy, Faculty of Medical Technology, Mahidol University, Salaya, Phuttamonthon, Nakhon Pathom 73170, Thailand; chutipong.suk@mahidol.ac.th; 4HydroSciences Montpellier (HSM), University Montpellier, CNRS, IRD, 34090 Montpellier, France; sylvie.manguin@ird.fr

**Keywords:** β-caryophyllene oxide, vetiver oil, excito-repellency, synergies, avoidance behavior, mosquito vectors

## Abstract

**Simple Summary:**

The deployment of synergistic insecticide combinations is intended to reduce both the dose of the substances used and the danger of insect populations developing physiological resistance to these insecticides. In order to test this hypothesis, two compounds (β-caryophyllene oxide and vetiver oil), with proven mosquito-repellent properties, were combined to enhance repellent efficacy. In general, the mixture of the compounds had a much stronger effect on mosquitoes than the individual compounds. The combination of β-caryophyllene oxide and vetiver oil produced additive contact irritability as a noncontact repellent, showing knockdown activities at low concentrations, indicating that combinations of these two repellent compounds can be used to develop a mosquito repellent that is more effective than a single compound. From a practical standpoint, both compounds must be formulated as herbal products and must undergo preliminary laboratory testing.

**Abstract:**

Repellents play a major role in reducing the risk of mosquito-borne diseases by preventing mosquito bites. The present study evaluated the mosquito-repellent activity of β-caryophyllene oxide 1% (BCO), vetiver oil 2.5% (VO), and their binary mixtures (BCO + VO (1:1), BCO + VO (2:1), BCO + VO (1:2)) against four laboratory-colonized mosquito species, *Aedes aegypti* (L.), *Aedes albopictus* (Skuse), *Anopheles minimus* Theobald, and *Culex quinquefasciatus* Say, using an excito-repellency assay system. In general, the compound mixtures produced a much stronger response in the mosquitoes than single compounds, regardless of the test conditions or species. The greatest synergetic effect was achieved with the combination of BCO + VO (1:2) in both contact and noncontact trials with *An. minimus* (74.07–78.18%) and *Cx. quinquefasciatus* (55.36–83.64%). Knockdown responses to the binary mixture of BCO + VO were observed for *Ae. albopictus*, *An. minimus,* and *Cx. quinquefasciatus*, in the range of 18.18–33.33%. The synergistic repellent activity of BCO and VO used in this study may support increased opportunities to develop safer alternatives to synthetic repellents for personal protection against mosquitoes.

## 1. Introduction

Mosquitoes are the deadliest animals on the planet given the number of particular disease pathogens they are capable of transmitting [1]. Their ability to carry and spread diseases to humans is responsible for millions of deaths every year. The main diseases are malaria, lymphatic filariasis, dengue, Japanese encephalitis, chikungunya, and Zika. *Aedes aegypti* (L.) and *Aedes albopictus* (Skuse) are the primary vectors of dengue and chikungunya viruses in Thailand. *Anopheles minimus* (Theobald) is one of the most important malarial vectors in forested areas of Thailand and other countries in the Greater Mekong Subregion. *Culex quinquefasciatus* Say is the most common vector across urban and semiurban areas and transmits *Wuchereria bancrofti*, *Plasmodium* (avian malaria), myxomatosis virus, encephalitis viruses, and other disease agents across the world [2]. Since vaccines are not currently available for most of these diseases, one of the most efficient methods of disease control is to rely on vector control approaches, as promoted by the World Health Organization (WHO), to reduce disease transmission [3].

Multiple measures, such as vaccination, preventive medications, and vector control, may be employed alone or in combination to limit the spread of diseases transmitted by vectors. Among these tools, vector control is an approach widely used to prevent or control outbreaks for most vector-borne diseases [4]. In general, mosquito control using chemical insecticides remains the most feasible technique for the control of many vectors. Unfortunately, many important mosquito species have developed resistance to different insecticides, especially *Ae. aegypti*, which has been reported to be resistant to many active ingredients across much of Thailand [5,6]. Therefore, one alternative method to combat blood-sucking insects is to use insect repellents as a form of personal protection to prevent transmission [7].

Mosquito repellents play a major role in disrupting disease transmission by reducing human–vector contact. DEET, previously called *N*,*N*-diethyl-m-toluamide or now *N*,*N*-3-methylbenzamide, was first registered in the USA in 1957, for use by military personnel in insect-infested areas. At present, DEET remains the gold standard insect repellent, a compound that is effective against most biting insects and other arthropod pests (such as mosquitoes, biting flies, ticks, fleas, and chigger mites) [7]. DEET has a remarkable safety profile after 66 years of extensive use, but toxic reactions have been reported in highly sensitive people or whenever the product is misapplied or misused [8].

Essential oils are complex mixtures of the volatile organic compounds present in plants. The repellent properties of several essential oils appear to be associated with specific compounds. Commercial applications of vetiver grass (*Vetiveria zizanioides* (L.) Nash:syn. *Chrysopogon zizanioides* (L.) Roberty) mostly involve the production of vetiver oil (VO) using root distillation. VO consists of a complex mixture of more than 300 compounds, with the major ones being vetiverol, vetivene, alpha- and beta-vetivone, khusimol, elemol, vetiselinenol, beta-eudesmol, terpenes, zizanoic acid, vanillin, hydrocarbons, sesquiterpenes, alcohols, and ketones [9]. Vetiver grass has been reported to have insect-repellent properties against ants, ticks, cockroaches, termites, mosquitoes, weevils, and beetles [10,11,12,13]. In addition to essential oils, a pure compound named β-caryophyllene oxide (BCO) is a bicyclic sesquiterpene, a representative of an epoxide derived from the olefin of (E)-caryophyllene. This compound is a common sesquiterpene present in many well-known aromatic-repellent plants, such as cloves, basil, cinnamon, and citrus [14]. A more recent study showed that BCO is an efficient mosquito repellent [15], and other studies showed that vetiver oil exhibits irritant and repellent activities against mosquitoes [16]. Furthermore, the phototoxicity and genotoxicity of BCO and VO were investigated by Nararak et al. in 2020 and 2022 [17,18]. According to the findings on phototoxic and genotoxic dangers, BCO and VO have no phototoxic potential and no substantial genotoxic response. As a result, these plant-based derivatives with repellent properties offer a viable alternative to synthetically created active compounds.

An excito-repellency (ER) test system has been used to evaluate the avoidance behavior of mosquitoes of test compounds [19,20,21]. This ER system allows the insecticide avoidance behavior of female mosquitoes to be studied by testing the contact irritancy and noncontact repellency of specific compounds. Contact irritability refers to direct tarsal contact with an insecticide that can cause a mosquito to escape from the test chamber. On the other hand, noncontact repellency results in insects detecting chemicals from a distance and escaping from the treated area without making physical contact with the insecticide.

Plant-based candidates, VO and BCO, were initially selected based on our previous studies [15,16,17,18]. A more recent study reported that BCO at 1% showed spatial repellency against *Ae. aegypti* (29.9%), *Ae. albopictus* (25.45%), *An. dirus* (31.67%), and *An. minimus* (86.9%), as well as high contact irritancy rates for *Ae. aegypti* (59.3%), *Ae. albopictus* (56.36%), *An. dirus* (32.73%), and *An. minimus* (92.2%) [15,17]. On the other hand, essential oil from VO at 2.5–5% elicited great repellency responses in *Ae. albopictus* (63.7%) and *An. minimus* (66.05%) [18,22]. Thus, VO and its two constituents (valencene and vetiverol) could be considered as safe repellents, and effective against mosquitoes [22]. Based on the spatial repellent activities previously shown by VO and BCO, the current study used both compounds to compare the behavioral responses of *Ae. aegypti*, *Ae. albopictus*, *An. minimus*, and *Cx. quinquefasciatus* to single and combined mixtures of VO and BCO using ER test chambers. 

## 2. Materials and Methods

### 2.1. Mosquito Populations

The laboratory strains used in this study were *Ae. aegypti* (USDA strain), *Ae. albopictus* (KU strain), *An. minimus* (KU strain), and *Cx. quinquefasciatus* (NIH strain). *Aedes aegypti* eggs were obtained from the U.S. Department of Agriculture (USDA), Gainesville, FL, USA. Samples of *Ae. albopictus* were originally captured in 1996 in Chanthaburi Province, eastern Thailand, by staff from the Ministry of Public Health, Thailand. The *An. minimus* colony originated from animal quarters in Rong Klnag district, Prae province, northern Thailand, in 1993, and has been maintained since then in the Department of Entomology, Faculty of Agriculture, Kasetsart University, Thailand. *Culex quinquefasciatus*, obtained from the National Institute of Health (NIH), Ministry of Public Health, Thailand, was originally collected from Pom Prap Sattru Phai, Bangkok, Thailand, in 1978 [23]. Specimens of these four species were reared in the insectary of the Department of Entomology, Faculty of Agriculture, Kasetsart University, Bangkok, Thailand, at 25 ± 5 °C and 80 ± 10% relative humidity with a 12 h:12 h light-to-dark photoperiod. Non-blood-fed females aged 5 days were used in all tests. Mosquitoes were deprived of a sugar meal for 24 h before testing but were provided with water-soaked cotton pads.

### 2.2. Test Compounds

Vetiver oil was purchased from the Thai-China Flavours and Fragrances Industry Co., Ltd. (Ayutthaya, Thailand). β-Caryophyllene oxide was purchased from the Acros Organics Company Ltd. (Morris Plains, NJ, USA). (95% purity, Lot No: A0356135). Based on our previous studies [16,22], the concentrations of VO at 2.5% and BCO at 1% were used to evaluate mosquito behavior. The five treatment combinations of VO (2.5%) and BCO (1%) are shown in Table 1.

### 2.3. Filter Paper Treatment

A mixture of VO and BCO was diluted with absolute ethanol to provide the concentrations mentioned in Table 1. Subsequently, 2.8 mL (253.75 cm^2^ per paper) of test solution was spread evenly over untreated filter paper (14.7 (width) × 17.5 (height) cm) (Whatman^®^ No. 1) using a 5 mL pipette and pipette controller, following the WHO procedure for testing papers with insecticidal compounds. Four similar sets of treated papers were prepared for each concentration, whereas control papers were treated in the same manner using only absolute ethanol. All treated papers were air-dried in a horizontal position at room temperature for 1 h before starting the test [15]. Multiple papers at each concentration were prepared, as each paper was used only once and then discarded.

### 2.4. Contact Irritancy and Noncontact Repellency Tests

An excito-repellency test system was used to evaluate irritancy and repellency responses of the mosquito vectors [24,25]. This system consisted of two treatment chambers containing repellent-treated papers (one chamber for the contact and the other for the noncontact treatment) and two matched control chambers containing only ethanol (nonrepellent)-treated papers. In the contact chambers, the four treated papers were placed in front of four inner screens, where mosquitoes made direct physical contact on the treated areas. For the noncontact configuration, all treated papers were placed behind the inner screens, where mosquitoes could not make physical contact with the treated surface.

All tests were performed between 0800 and 1600 h [26]. Fifteen starved female mosquitoes were introduced into each test chamber and exposed to the environmental conditions inside the chamber for 3 min. Then, the exit door of each test chamber was opened to allow mosquitoes to exit from the test or control chamber to a receiving paper box connected to the chamber. Escaped mosquitoes were recorded at 1 min intervals up until 30 min. After the exposure time, escaping and remaining mosquitoes were separately removed from the chambers and kept in clean plastic holding cups, provided with cotton pads soaked with 10% sugar solution for 24 h. Knockdown after the 30 min exposure time and mortality at 24 h were noted. Four replicates were performed for each concentration, treatment, and control.

### 2.5. Data Analysis

Before analysis, each escape percentage was adjusted based on the number of paired control escape responses using Abbott’s formula [27]. The Kaplan–Meier survival analysis method was used to analyze and interpret mosquito behavioral response data. Multiple log-rank tests were used to compare two Kaplan–Meier survival curves for escape response data of mosquitoes exposed to contact and noncontact chambers, control, and treatment, concentrations, substances and species, using the SAS version 9 software package (SAS Institute, Cary, NC, USA) [19]. Survival curve comparison patterns were considered significantly different at *p* < 0.05. Escape times (ETs) in minutes were recorded from the beginning of the test for 25% (ET_25_), 50% (ET_50_), and 75% (ET_75_) of the test mosquitoes to escape from the test chamber. 

## 3. Results

The escape responses of the four mosquito species were tested using exposure to a single chemical or dual mixture of chemical repellents, specifically BCO 1% and VO 2.5% (Table 1). The escape patterns during the 30-min exposure period for the four mosquito species are given in Figure 1, Figure 2, Figure 3 and Figure 4. The escape rates represent the probabilities for mosquitoes escaping from a chamber for a particular chemical or mixture of chemicals and concentration. The percentages of the populations of the four species that escaped within a 30 min period of exposure to either an individual or a combination of BCO and VO in the contact and noncontact trials are presented in Table 2. The percentage represents the probability of mosquitoes escaping from the chamber with a given chemical repellent and formulation. Strong escape responses were observed with mixtures of BCO + VO (1:1) in both the noncontact and contact trials for *An. minimus* (61.54–80.39%) and *Cx. quinquefasciatus* (51.85–75.47%), as well as for BCO + VO (1:2) for *An. minimus* (74.07–78.18%) and *Cx. quinquefasciatus* (55.36–83.64%) (Table 2, Figure 3 and Figure 4). For *Aedes species*, a higher percentage of escaped mosquitoes was observed with BCO + VO (2:1) in the contact and noncontact trials for *Ae. aegypti* (40.35–53.70%) and for contact only for *Ae. albopictus* (67.27%). For BCO + VO (1:1), the latter species had high escape rates in the contact and noncontacts trials (53.85–62.75%), as shown in Table 2 and Figure 1 and Figure 2. Overall, *An. minimus* and *Cx. quinquefasciatus* demonstrated more robust escape responses than *Ae. aegypti* and *Ae. albopictus*. With *Ae. albopictus*, a weak noncontact escape pattern was found for BCO + VO (2:1), as shown in Table 2 and Figure 3 and Figure 4.

The knockdown of escaped and nonescaped mosquitoes was observed during the exposure period (30 min), as shown in Table 3. The percentage knockdown of nonescaped *Ae. albopictus* specimens in the contact trials for BCO + VO (1:1) was 33.33%, 23.8% for BCO + VO (1:2), and 22.22% for BCO + VO (2:1). For *An. minimus* and *Cx. quinquefasciatus*, the percentage knockdown of nonescaped specimens in the contact trials for BCO + VO (2:1) were 29.41% and 18.18%, respectively (Table 3). No knockdown specimens were observed for *Ae. aegypti*. Overall, there was no mortality in any of the test populations when exposed to either plant-based repellent (Table 3). 

The multiple log-rank comparisons between two species exposed to repellent compounds in either the contact or noncontact trials are shown in Table 4. For the contact trials, there were significant differences (*p* < 0.05) in the escape responses when *Ae. aegypti* was compared with the other species at the two concentrations of BCO + VO (1:1 and 1:2) but not for BCO + VO (2:1). For the noncontact trials, significant differences (*p* < 0.05) in the escape responses between two species were found for most pairs for BCO + VO (1:2) (Table 4). Table 5 presents the log-rank comparisons of the mosquito escape responses between paired concentrations of BCO and VO in the contact and noncontact trials. There were no significant differences in either paired contact or noncontact trials in *Cx. quinquefasciatus* for all comparisons (*p* > 0.05). For *Ae. aegypti*, significant differences were found for BCO + VO (2:1) vs. BCO + VO (1:2) and BCO + VO (2:1) vs. BCO + VO (1:1) in contact trials. Significant differences were also found in four cases of *Ae. albopictus* in a noncontact trial (BCO vs. BCO + VO (2:1), VO vs. BCO + VO (2:1), BCO + VO (2:1) vs. BCO + VO (1:2), and BCO + VO (2:1) vs. BCO + VO (1:1)). For *An. minimus*, differences were significant in three cases, including BCO vs. BCO + VO (1:2), VO vs. BCO + VO (1:2), and VO vs. BCO + VO (1:2) in noncontact trials. The BCO + VO (1:2) vs. BCO + VO (1:1) comparison was significant in both contact and noncontact trials with *An. minimus*, respectively (Table 5). Multiple log-rank comparisons were conducted between the contact and noncontact trials for each single compound or mixture (Table 6). Interestingly, no significant differences in escape patterns for *Ae. aegypti* were observed in the paired comparisons between the contact and noncontact trials. For *Ae. albopictus*, no significant differences in escape patterns were apparent in the paired comparisons between the contact and noncontact trials, except for BCO + VO (2:1). Likewise, marked differences in the escape responses were found in the paired contact and noncontact trials (*p* < 0.05), with BCO + VO (1:1) for *An. minimus*. Escape responses were also significantly different between the contact and noncontact trials when *Cx. quinquefasciatus* was tested against BCO, VO, and BCO + VO (1:2), as shown in Table 6.

The escape patterns of the mosquitoes from chambers treated with test compounds were defined as escape times for 25% (ET_25_), 50% (ET_50_), and 75% (ET_75_) of the test populations to leave the treated chambers within 30 min (Table 7). *Aedes albopictus* had a faster response to all single or combinations of BCO and VO compounds with ET_25_ values of 1–2 min in both contact and noncontact trials. *Aedes aegypti* had delayed escape responses to all combinations with ET_25_ values of 3–10 min in both contact and noncontact trials, except for BCO + VO (2:1), for which lower values of 2–3 min were recorded, respectively. For ET_75_, no response was observed in either contact or noncontact trails for *Ae. aegypti* and *Ae. albopictus* for all test compounds. *Anopheles minimus* had fast escape responses of 1–3 min at ET_25_–ET_50_ to BCO + VO (1:2) in both contact and noncontact trials and at ET_25_ for all compounds, except for BCO + VO (1:1). *Culex quinquefasciatus* displayed ET_25_ values of 1–3 min in contact trials for all compounds tested but showed a delayed response of 3–6 min in noncontact trials (Table 7).

## 4. Discussion

Currently, insecticide resistance poses a serious threat to the success of vector control programs and has added to the pressure on scientists to develop new and enhanced vector control tools [28]. Globally, numerous studies, including the present investigation, have clearly identified that traditionally used plant-based insect repellents are promising and could control arthropod pests and vectors transmitting disease agents [29]. Repellents play an important role in preventing or reducing the incidence of vector-borne diseases by preventing human–vector contact. Synthetic repellents are a conventional means of personal protection against most biting insects and pests, with the worldwide market for personal insect repellents estimated at more than USD 2 billion annually [30]. One of the most effective and widely used insect repellents is *N, N*-diethyl-meta-toluamide (DEET) [31]. There are more than 200 different DEET products available commercially, ranging from concentrations of 5 to 100% [14]. However, critical concerns have been raised over the safety of DEET in other studies [32,33,34].

In general, plant-based repellents are known to be safer and environmentally friendlier alternative sources compared with chemical insect repellents. The plant products that are in use include a wide range of substances, from crude plant extracts to essential oils and isolated compounds. In the present study, the combination of BCO and VO showed repellent efficacy against laboratory-colonized mosquito populations. VO has been reported to show a high repellent efficacy against mosquitoes [16,22]. The repellent properties of some plant-based repellents against many arthropods are based on their aromatic constituents [35]. BCO is widely found in plant-based essential oils and presents the capacity to repel mosquitoes [36,37,38,39,40,41,42,43]. More recent studies have reported that *Ae. aegypti*, *Ae. albopictus*, *An. minimus,* and *An. dirus* had high avoidance response rates to 1% concentrations of BCO compared with DEET at the same concentration [15,20], showing the high repellency potential of this compound. 

Deploying synergistic combinations aims to reduce the dose of the substances and to help with reducing the risk of developing physiological resistance by any insect population. To support this approach, we mixed two compounds with confirmed mosquito repellent activity to increase repellent efficacy—three binary mixture forms of BCO (1%) and VO (2.5%) were prepared to test the excito-repellency activity on *Ae. aegypti*, *Ae. albopictus*, *An. minimus,* and *Cx. quinquefasciatus* using an ER repellency test system. The combinations of BCO and VO (2:1) produced higher escape percentages in contact trials with *Aedes* mosquitoes than a single compound. Similar to the preceding investigation, Panthawong et al. [44] documented the observation of robust escape behavior in contact trials involving VO and *Andrographis paniculata* (AP). This escape pattern was consistently observed across all combinations, with escape percentages ranging from 56% to 73%. Notably, the combination ratio of 1:4 yielded the highest escape rate at 73.33% for VO: AP in the contact trial. Furthermore, Boonyuan et al. [45] reported that the most potent contact irritant effect was achieved with a VO: AP ratio of 1:1 (*v*:*v*) against *Cx. quinquefasciatus* mosquitoes, resulting in an impressive 96.67% escape rate, surpassing the effectiveness of DEET, which achieved an 88.67% escape rate.

Tisgratog et al. [46] conducted a comprehensive review, revealing that *An. minimus* and *Cx. quinquefasciatus* mosquitoes exhibited a heightened avoidance response to various essential oils, including *Cymbopogon nardus*, *Citrus hystrix*, *Nepeta cataria*, *Syzygium aromaticum*, and *Ocimum americanum*, when compared with *Aedes* mosquitoes. It is worth noting that the escape percentage observed could have been influenced by the specific mosquito species under investigation. *Anopheles* and *Culex* mosquitoes demonstrated a greater sensitivity to repellents than *Aedes* mosquitoes, as the latter are known to possess a higher tolerance to repellent substances [47]. In an additional study conducted Noosidum et al. (2014) [48], a mixture of *Litsea cubeba* (LC) and *Litsea salicifolia* (LS) at 0.075% against *Ae. aegypti* showed the highest synergistic action (65.5% escaped) compared with that with unmixed oil alone at the same concentration (LC = 20% and LS = 32.2%). In addition, mixtures of LC and LS at 0.075% demonstrated the highest noncontact repellency (62.7%), which was significantly better than the use of LC (20%) or LS (20.3%) alone. In a study of the protection period against mosquito bites, *Litsea* (*L. cubeba*) + rosewood (*Aniba rosaeodora*) in a ratio of 1:1 (*v*/*v*) at a 10% concentration showed 86% repellency for 4 h against *Ae. aegypti* using human-skin test methods (K & D module) [49]. However, some combinations produced a reduction (antagonistic effect) compared with the pure compound. Pavela (2014) [50] reported that L-carvone and gallic acid created an antagonistic effect with the other aromatic compounds to *Spodoptera littoralis* Boisd., a lepidopteran pest of crops. This suggested that the synergistic or additive effects of dual mixtures and the activity could be species-specific; therefore, no generalization to other insect species could be drawn from this study [51].

The combinations of BCO and VO were effective against the four tested mosquito vector species, especially *Cx. quinquefasciatus* and *An. minimus*, but some combinations were less effective when tested against *Ae. aegypti* and *Ae. albopictus*. However, the BCO and VO mixture provided better protection against mosquito bites than when applying a repellent alone. *Culex quinquefasciatus* exhibited much stronger escape responses against the mixtures of BCO: VO than other species. Phasomkusolsil and Soonwera (2011) [52] found that *An. minimus* and *Cx. quinquefasciatus* were more sensitive to several different oils compared with *Ae. aegypti*. Sathantriphop et al. (2014) [21] reported that *Cx. quinquefasciatus* and *An. minimus* had much stronger behavioral responses to VO than *Ae. aegypti* and *Ae. albopictus*. One study examining the effect of a repellent on the olfactory system of *Culex* mosquito antennal sensilla neurons showed that β-caryophyllene and (-)-caryophyllene oxide produced a strong response on short, blunt-tipped type I (SBT-I-B) sensilla [53].

In terms of toxicity, binary mixtures of BCO and VO produced a percentage knockdown at 1 h in a nonescape chamber. Notably, BCO + VO (2:1) produced knockdown activity with *Ae. albopictus*, *An. minimus,* and *Cx. quinquefasciatus*. Noosidum et al. (2014) [48] also tested ER with *L. cubeba* 0.175% and *L. salicifolia* 0.175% against *Ae. aegypti*. The synergist effect produced 67.8% knockdown. Another study presented a comparison of the synergistic effects of *Rosmarinus officinalis* L. essential oil constituents against the larvae and an ovarian cell line of the cabbage looper, *Trichoplusia ni* (Lepidoptera: Noctuidae). The insecticidal activity of rosemary oil appeared to be a consequence of the synergistic reaction between 1,8-cineole and (±) camphor [54].

## 5. Conclusions

In summary, the binary combination of BCO and VO demonstrated the ability to generate additive contact irritancy, noncontact repellency, and knockdown activity at low concentrations. These findings suggest that utilizing combinations of these two repellent compounds offers a more effective mosquito repellent solution compared with using either compound in isolation. The synergistic repellent activity observed in the essential oils employed in this study presents a promising alternative to synthetic repellents for personal protection against mosquitoes. Adopting such an approach holds the potential to alleviate the environmental burden associated with chemical repellents and promote the sustainable utilization of locally available bioresources within rural communities. Furthermore, future studies should focus on optimizing the efficacy of these repellents through the incorporation of potentiating agents. Additionally, there is a need for bioefficacy evaluations of these repellent combinations in field conditions. These studies will facilitate the translation of laboratory findings into practical applications, advancing our understanding of mosquito repellency and its impact on public health.

## Figures and Tables

**Figure 1 insects-14-00773-f001:**
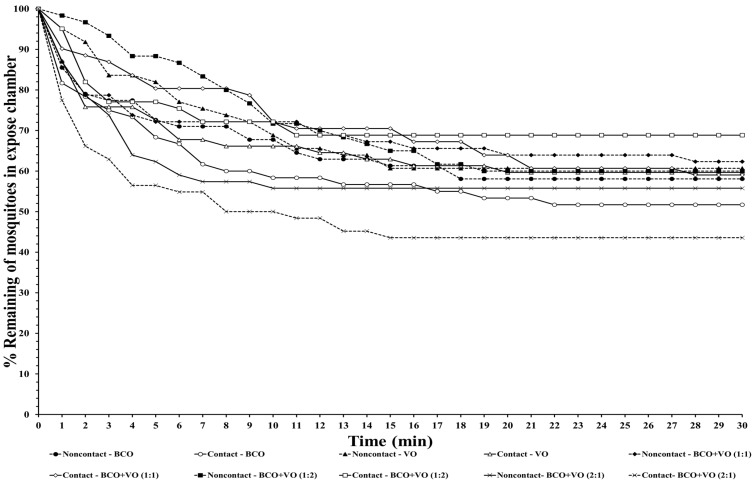
Escape curves for *Ae. aegypti* in treated noncontact and contact excito-repellency assays. Escape responses were recorded in 1 min intervals during exposure for 30 min to various combinations of and concentrations of β-caryophyllene oxide (BCO) and vetiver oil (VO). Paired control escape responses not shown.

**Figure 2 insects-14-00773-f002:**
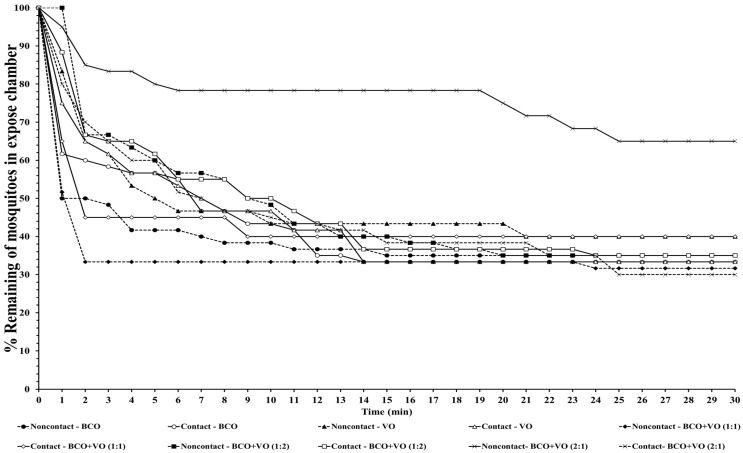
Escape curves for *Ae. albopictus* in treated noncontact and contact excito-repellency assays. Escape responses were recorded in 1 min intervals during exposure for 30 min to various combinations of concentrations of β-caryophyllene oxide (BCO) and vetiver oil (VO). Paired control escape responses not shown.

**Figure 3 insects-14-00773-f003:**
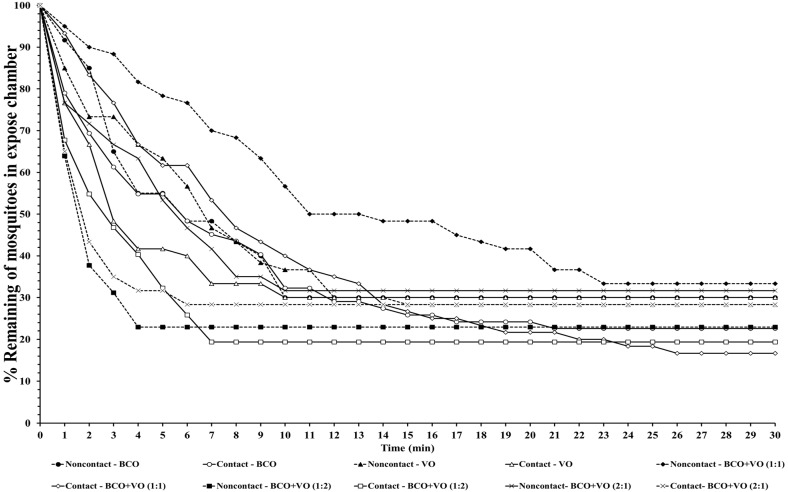
Escape curves for *An. minimus* in treated noncontact and contact excito-repellency assays. Escape responses were recorded in 1 min intervals during exposure for 30 min to various combinations of concentrations of β-caryophyllene oxide (BCO) and vetiver oil (VO). Paired control escape responses not shown.

**Figure 4 insects-14-00773-f004:**
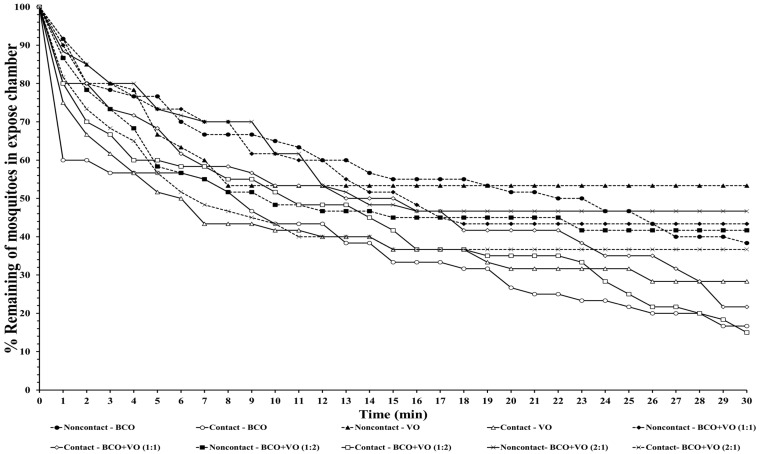
Escape curves for *Cx. quinquefasciatus* in treated noncontact and contact excito-repellency assays. Escape responses were recorded in 1 min intervals during exposure for 30 min to various combinations of concentrations of β-caryophyllene oxide (BCO) and vetiver oil (VO). Paired control escape responses not shown.

**Table 1 insects-14-00773-t001:** Treatment and combinations of β-caryophyllene oxide (BCO) and vetiver oil (VO).

Treatment and Treatment Combinations	Ratio	Symbol
β-caryophyllene oxide 1%	1	BCO
vetiver oil 2.5%	1	VO
β-caryophyllene oxide 1%: vetiver oil 2.5%	1:1	BCO + VO (1:1)
β-caryophyllene oxide 1%: vetiver oil 2.5%	2:1	BCO + VO (2:1)
β-caryophyllene oxide 1%: vetiver oil 2.5%	1:2	BCO + VO (1:2)

**Table 2 insects-14-00773-t002:** Percentage escape responses of mosquitoes exposed to one compound or a combination of β-caryophyllene oxide (BCO) and vetiver oil (VO).

Compound	Test Condition	*Ae. aegypti*		*Ae. albopictus*		*An. minimus*		*Cx. quinquefasciatus*	
		N	%Esc * (N)	N	%Esc (N)	N	%Esc (N)	N	%Esc (N)
BCO	TC	60	44.64 (29)	60	60.00 (40)	62	75.00 (48)	60	81.48 (50)
	CC	60	6.67 (4)	60	16.67 (10)	61	8.20 (5)	60	10.00 (6)
	TN	62	37.04 (26)	60	59.62 (39)	60	67.27 (42)	60	58.18 (37)
	CN	60	10.00 (6)	60	13.33 (8)	59	8.47 (5)	60	8.33 (5)
VO	TC	62	35.19 (25)	60	60.78 (40)	60	66.67 (42)	60	71.19 (43)
	CC	62	9.68 (6)	60	15.00 (9)	60	10.00 (6)	60	1.67 (1)
	TN	61	35.17 (24)	60	55.56 (36)	60	68.52 (43)	60	39.62 (28)
	CN	61	6.56 (4)	60	10.00 (6)	60	10.00 (6)	60	11.67 (7)
BCO + VO (1:1)	TC	61	25.19 (25)	60	53.85 (36)	60	80.39 (50)	60	75.47 (47)
	CC	61	9.84 (6)	60	13.33 (8)	60	15.00 (9)	60	11.67 (7)
	TN	61	27.45 (23)	60	62.75 (41)	60	61.54 (40)	60	51.85 (34)
	CN	61	14.75 (9)	60	15.00 (9)	60	13.33 (8)	60	10.00 (6)
BCO + VO (2:1)	TC	62	53.70 (35)	60	67.27 (42)	60	67.92 (43)	60	60.71 (38)
	CC	62	9.68 (6)	60	8.33 (5)	60	11.67 (7)	60	6.67 (4)
	TN	61	40.35 (27)	60	29.09 (21)	60	64.81 (41)	60	50.88 (32)
	CN	61	4.92 (3)	60	8.33 (5)	60	9.84 (6)	60	4.92 (3)
BCO + VO (1:2)	TC	61	31.15 (19)	60	60.38 (39)	62	78.18 (50)	60	83.64 (51)
	CC	60	0	60	11.67 (7)	60	8.33 (5)	60	8.33 (5)
	TN	59	37.29 (22)	60	58.00 (39)	61	74.07 (47)	60	55.36 (35)
	CN	60	1.67 (1)	60	16.67 (10)	59	8.47 (5)	60	6.67 (4)

* Treatment percentage escape adjusted based on paired control responses. BCO: β-caryophyllene oxide, VO: vetiver oil, Esc: escape mosquitoes, N: number, TC: treatment contact, CC: contact control, TN: treatment noncontact. CN: control noncontact.

**Table 3 insects-14-00773-t003:** Percentage knockdown per mosquito species exposed to one compound or a combination of β-caryophyllene oxide (BCO) and vetiver oil (VO).

	Test Condition	% Knockdown (30 min)							
		*Ae. aegypti*		*Ae. albopictus*		*An. minimus*		*Cx. quinquefasciatus*	
		Es	NEs	Es	NEs	Es	NEs	Es	NEs
BCO	TC	-	-	-	-	-	-	-	-
	CC	-	-	-	-	-	-	-	-
	TN	-	-	-	-	-	-	-	-
	CN	-	-	-	-	-	-	-	-
VO	TC	-	-	-	-	-	-	-	-
	CC	-	-	-	-	-	-	-	-
	TN	-	-	-	-	-	-	-	-
	CN	-	-	-	-	-	-	-	-
BCO + VO (1:1)	TC	-	-	-	33.33	-	-	-	-
	CC	-	-	-	-	-	-	-	-
	TN	-	-	-	-	-	-	-	-
	CN	-	-	-	-	-	-	-	-
BCO + VO (2:1)	TC	-	-	-	22.22	-	29.41	-	18.18
	CC	-	-	-	-	-	-	-	-
	TN	-	-	-	-	-	-	-	-
	CN	-	-	-	-	-	-	-	-
BCO + VO (1:2)	TC	-	-	-	23.8	-	-	-	-
	CC	-	-	-	-	-	-	-	-
	TN	-	-	-	-	-	-	-	-
	CN	-	-	-	-	-	-	-	-

BCO: β-caryophyllene oxide, VO: vetiver oil, Es: escaped mosquitoes, NEs: nonescaped mosquitoes, N: Number, TC: treatment contact, CC: contact control, TN: treatment non-contact. CN: control non-contact.

**Table 4 insects-14-00773-t004:** Comparisons of mosquito escape responses between contact and noncontact chambers for mosquito species exposed to β-caryophyllene oxide (BCO) and vetiver oil (VO).

Compound	Mosquito Species	Contact Treatment	Noncontact Treatment
	*Ae. aegypti* vs. *Ae. albopictus*	0.035	0.0031
	*Ae. aegypti* vs. *An. minimus*	0.0017	0.0028
BCO	*Ae. aegypti* vs. *Cx. quinquefasciatus*	0.0004	0.0945
	*Ae. albopictus* vs. *An. minimus*	0.3834	0.6156
	*Ae. albopictus* vs. *Cx. quinquefasciatus*	0.2308	0.1395
	*An. minimus* vs. *Cx. quinquefasciatus*	0.8996	0.099
	*Ae. aegypti* vs. *Ae. albopictus*	0.008	0.0064
	*Ae. aegypti* vs. *An. minimus*	0.0017	0.0002
VO	*Ae. aegypti* vs. *Cx. quinquefasciatus*	0.0018	0.2848
	*Ae. albopictus* vs. *An. minimus*	0.4895	0.5086
	*Ae. albopictus* vs. *Cx. quinquefasciatus*	0.6846	0.0734
	*An. minimus* vs. *Cx. quinquefasciatus*	0.7974	0.0159
BCO + VO (1:1)	*Ae. aegypti* vs. *Ae. albopictus*	0.006	0.0001
	*Ae. aegypti* vs. *An. minimus*	<0.0001	0.008
	*Ae. aegypti* vs. *Cx. quinquefasciatus*	<0.0001	0.0682
	*Ae. albopictus* vs. *An. minimus*	0.3286	0.06
	*Ae. albopictus* vs. *Cx. quinquefasciatus*	0.4551	0.0153
	*An. minimus* vs. *Cx. quinquefasciatus*	0.1771	0.4685
BCO + VO (2:1)	*Ae. aegypti* vs. *Ae. albopictus*	0.2884	0.1919
	*Ae. aegypti* vs. *An. minimus*	0.0289	0.0203
	*Ae. aegypti* vs. *Cx. quinquefasciatus*	0.6204	0.6583
	*Ae. albopictus* vs. *An. minimus*	0.1802	<0.0001
	*Ae. albopictus* vs. *Cx. quinquefasciatus*	0.6284	0.0336
	*An. minimus* vs. *Cx. quinquefasciatus*	0.0525	0.0192
BCO + VO (1:2)	*Ae. aegypti* vs. *Ae. albopictus*	0.0004	0.0005
	*Ae. aegypti* vs. *An. minimus*	0.0001	<0.0001
	*Ae. aegypti* vs. *Cx. quinquefasciatus*	<0.0001	0.0035
	*Ae. albopictus* vs. *An. minimus*	0.0044	0.0016
	*Ae. albopictus* vs. *Cx. quinquefasciatus*	0.1233	0.6618
	*An. minimus* vs. *Cx. quinquefasciatus*	0.1706	0.0007

**Table 5 insects-14-00773-t005:** Log-rank comparisons of escape responses between combinations of β-caryophyllene oxide (BCO) and vetiver oil (VO) of four mosquito species.

Mosquito Species	Test Compounds (Ratio)	Contact Treatment	Non-Contact Treatment
*Ae. aegypti*	BCO vs. VO	0.3984	0.6613
	BCO vs. BCO + VO (2:1)	0.2912	0.6761
	BCO vs. BCO + VO (1:2)	0.0603	0.3137
	BCO vs. BCO + VO (1:1)	0.275	0.6419
	VO vs. BCO + VO (2:1)	0.0588	0.3449
	VO vs. BCO + VO (1:2)	0.2783	0.5343
	VO vs. BCO + VO (1:1)	0.8499	0.9767
	BCO + VO (2:1) vs. BCO + VO (1:2)	0.0038	0.1189
	BCO + VO (2:1) vs. BCO + VO (1:1)	0.0284	0.4267
	BCO + VO (1:2) vs. BCO + VO (1:1)	0.3798	0.5955
*Ae. albopictus*	BCO vs. VO	0.67	0.2742
	BCO vs. BCO + VO (2:1)	0.9747	0.0001
	BCO vs. BCO + VO (1:2)	0.4986	0.186
	BCO vs. BCO + VO (1:1)	0.661	0.6579
	VO vs. BCO + VO (2:1)	0.7352	0.002
	VO vs. BCO + VO (1:2)	0.8179	0.8971
	VO vs. BCO + VO (1:1)	0.9292	0.0654
	BCO + VO (2:1) vs. BCO + VO (1:2)	0.5881	0.0006
	BCO + VO (2:1) vs. BCO + VO (1:1)	0.8006	<0.0001
	BCO + VO (1:2) vs. BCO + VO (1:1)	0.6916	0.0381
*An. minimus*	BCO vs. VO	0.6909	0.9724
	BCO vs. BCO + VO (2:1)	0.4881	0.8315
	BCO vs. BCO + VO (1:2)	0.1166	0.0055
	BCO vs. BCO + VO (1:1)	0.826	0.1207
	VO vs. BCO + VO (2:1)	0.2122	0.8474
	VO vs. BCO + VO (1:2)	0.0952	0.0173
	VO vs. BCO + VO (1:1)	0.8674	0.0137
	BCO + VO (2:1) vs. BCO + VO (1:2)	0.7613	0.1218
	BCO + VO (2:1) vs. BCO + VO (1:1)	0.3607	0.1208
	BCO + VO (1:2) vs. BCO + VO (1:1)	0.0485	0.0005
*Cx. quinquefasciatus*	BCO vs. VO	0.2085	0.2843
	BCO vs. BCO + VO (2:1)	0.0863	0.5878
	BCO vs. BCO + VO (1:2)	0.6457	0.7996
	BCO vs. BCO + VO (1:1)	0.1946	0.8176
	VO vs. BCO + VO (2:1)	0.555	0.7484
	VO vs. BCO + VO (1:2)	0.3195	0.212
	VO vs. BCO + VO (1:1)	0.9356	0.5262
	BCO + VO (2:1) vs. BCO + VO (1:2)	0.1285	0.3755
	BCO + VO (2:1) vs. BCO + VO (1:1)	0.5108	0.7984
	BCO + VO (1:2) vs. BCO + VO (1:1)	0.3062	0.5689

**Table 6 insects-14-00773-t006:** Comparisons of mosquito escape responses between contact and noncontact chambers for *Ae. aegypti*, *Ae. albopictus*, *An. minimus,* and *Cx. quinquefasciatus* exposed to β-caryophyllene oxide (BCO) and vetiver oil (VO).

Compound	Mosquito Species			
	*Ae*. *aegypti*	*Ae*. *albopictus*	*An*. *minimus*	*Cx*. *quinquefasciatus*
BCO	0.467	0.8585	0.3947	0.0029
VO	0.733	0.7372	0.6486	0.0115
BCO + VO (1:1)	0.865	0.2418	0.0145	0.0539
BCO + VO (2:1)	0.1742	<0.0001	0.1484	0.1137
BCO + VO (1:2)	0.9072	0.8372	0.6154	0.0216

**Table 7 insects-14-00773-t007:** Estimation of the escape time in minutes for 25% (ET_25_), 50% (ET_50_), and 75% (ET_75_) of mosquitoes exposed to serial concentrations of β-caryophyllene oxide (BCO) and vetiver oil (VO) in contact and noncontact chambers.

Mosquito Species	Test	VO			BCO			BCO + VO (1:1)			BCO + VO (2:1)			BCO + VO (1:2)		
		ET_25_	ET_50_	ET_75_	ET_25_	ET_50_	ET_75_	ET_25_	ET_50_	ET_75_	ET_25_	ET_50_	ET_75_	ET_25_	ET_50_	ET_75_
*Ae. aegypti*	C	5	-	-	3	-	-	10	-	-	2	8	-	7	-	-
	NC	8	-	-	5	-	-	4	-	-	3	-	-	10	-	-
*Ae. albopictus*	C	1	7	-	1	7	-	1	2	-	1	2	-	2	9	-
	NC	2	5	-	1	-	-	1	-	-	1	-	-	2	9	-
*An. minimus*	C	2	3	-	2	6	17	4	8	16	1	2	-	1	3	7
	NC	2	7	-	3	6	-	7	11	-	2	6	-	1	2	4
*Cx. quinquefasciatus*	C	1	6	-	1	9	21	3	13	29	2	7	-	2	11	25
	NC	5	-	-	6	22	-	5	16	-	5	14	-	3	10	-

C = contact; NC = noncontact.

## Data Availability

All relevant data are included in the article.

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
