# Peer review of "Synergistic Repellent and Irritant Effects of a Mixture of β-Caryophyllene Oxide and Vetiver Oil against Mosquito Vectors"

_insects, 2023, doi:10.3390/insects14090773_

Round 1

Reviewer 1 Report

The present article by Nararak et al., on repellency and irritant properties of binary mixture of BCO and VO is reasonably well written and presented just fine. There are some major issues that need to be addressed before the MS becomes eligible for publication in the present journal. 

The abstract, intro, M&M and results are well written and presented (please see below for specific comment son each section). But the discussion and conclusion could be improved significantly to make the MS more relevant and a good read. Those two sections (discussion and conclusion) need some language and content improvement. I will detail them as much as possible below:

1.     The introduction is overall very well written, one major thing that is missing in my opinion is environmental impact of using these plant-based alternatives. i.e., just mentioning that since these are plant based, they are better for the environment is not enough.

2.     This is not something that is required, but just checking out: why no recently colonized or even field collected mosquitoes were used? Also, the strains used in this study has any resistance to any known synthetic or other chemicals presently used for their control/repellency? If yes, adding that information (mainly in discussion section) and discussing your experiments in that context would make the present MS better.

3.     Line 140: please provide more info on the filter paper and the measurements mentioned in the brackets makes no sense. Is it its radius or diameter? Please clarify.

4.     Is extracted VO and BCO by themselves volatile? Clarify that and if yes, does that have any effect/relevance to the present study.

5.     Line 160: why such a large window is being used to perform the experiment? Also, would it be better to perform these experiments when these different species are most active (based on their circadian rhythm) which generally correspond to their biting behavior? Please clarify.

6.     Line 168: use alternates to word “required” as you have already performed these experiments and reporting it here.

7.     In results when presenting a range of response, either use +SD/SEM or inf you want to present it as a range it must be present from lower % to higher % (at least that’s the general practice).

8.     Overall, the results are well written. But I would recommend that since so much data is being presented in 4 or 5 paragraphs, take care to make sure fig/table references are accurate. I myself didn’t notice any discrepancies, except that sometimes the first reference to figures are not in the order of their appearance/numbering.

9.     Line 271: plant-based repellent to plant-based repellent alone.

10.  The main issue with the present MS is the discussion part. It not only needs many modifications w.r.t. language (some I will note below), but also regarding content that is being discussed. For ex, paragraphs from results (line 264-272) is not well discussed at all, instead there are many references to information that could effectively go into introduction (like line 342-345 on DEET). Overall, the discussion could be made more relevant by discussing your data in the light of other studies in the field and relevancy of your experiments and data to the improvement of the overall question (in this case having new/novel repellents).

11.  Line 339: man=human

12.  Line 362-363: starting at “the combination of…” is this statement accurate to all the 4 species you have tested here?

13.  Some of the binomial nomenclatures (mainly in discussion section) is not italicized, please check them.

14.  Line 386: either use vetiver oil or VO consistently instead simply of vetiver like at the start of this line.

15.  Line 389: blunt-tipped type-1 (SBT-I-B) neuron= blunt-tipped type-1 (SBT-I-B) sensilla

16.  Conclusion: this section could definitely modified to read a bit better than it is now. I have couple suggestions below.

17.  Line 403: one only=use the term alone. 

18.  Line 404: delete for

19.  Line 404: alternatives= alternative.

check above.

Author Response

Thank you very much for your comments. We greatly appreciate your advise. For our response please see the attachment

Reviewer 2 Report

Line 74, you mention the remarkable safety profile of DEET after 40 years; actually based on your cited dates it should be 66 yrs.

Line 40 the dimensions of the filter paper need clarification.

Page 11, lines 290-304 there are several points that need clarification.  You say that significant differences were found in two cases in the non-contact trial of Ae. albopictus, but Table 5 indicates 5. For An. minimus you state that there are two significant differences for the non-contact trials, but Table 5 indicates 4.  Am I reading the table incorrectly? 

Author Response

We would like to express our sincere thanks to the reviewer for all the valuable comments. For our response please see the attachment. 

Round 2

Reviewer 1 Report

The authors did a commendable job with this version of the MS, it has improved very significantly and is almost ready for publication. I just have a couple comments/suggestions:

1. Line 93: I believe there should be "et al" after Nararak

2. Line 172: provided=performed.

3. In figures 1-4, while you sued the survival curve method to analyze and plot your data, you don't have to call it as such in your figure legends, that could be misleading to the reader as it is not a "survival curve". You can simply rephrase it as "Graph/curve percent of remaining mosquitoes in expose chamber" or something like that. 

Author Response

We greatly appreciate your efforts to enhance the MS. Please see our revised in the attachment.     
